# **Evaluating Yawed Turbine Transfer Functions from SCADA Data**

Aidan Gettemy<sup>1</sup>, Luke Abbatessa<sup>2</sup>, and Nathan L. Post<sup>3</sup>

<sup>1</sup>School of Natural Sciences and Mathematics, UT Dallas, TX, USA

**Correspondence:** Nathan L. Post (nathan@windtester.com)

Abstract. While nacelle transfer functions (NTFs) have been applied to correct free-stream wind speed measurements at the nacelle of turbines steered into the wind, less is known about the relationship between unsteered (non-yawed) and steered (yawed) wind measurements on turbines performing wake steering. As wake steering becomes an important tool for maximizing collective wind farm power, determining and correcting bias caused by prolonged yaw misalignment on wind measurements is critical to improving collective wind farm control and analysis. We propose a new approach for evaluating NTFs using SCADA data. Using SCADA and wake steering controller 1-minute statistics recorded over 3.5 months at a large utility-scale wind plant, we apply several consensus methods to estimate unsteered turbine measurements for steered turbines using neighboring turbines. Then a bagged tree regressor algorithm is trained to predict the unsteered wind direction, wind speed, and generator power using the measured SCADA data during wake steering using the best consensus estimate as the target value. With the NTFs estimated through the ML model, we define experimentally determined non-linear sensor bias in the measured data as a function of yaw angle.

#### 1 Introduction

Wake steering, the intentional yaw misalignment of upwind wind turbines to decrease wake interaction with downwind turbines, has been developed to improve overall wind farm energy generation. Wake interactions can reduce AEP by 10-20% Hansen et al. (2012). Wake steering enables wind farms to contain a denser grid of turbines, increasing the energy captured for the same cost of site development Fleming et al. (2017). During wake steering, wind turbines spend time at significant yaw angles relative to the direction of the wind. This deliberate and sustained misalignment can result in incoming wind direction and wind speed measurement errors that could impact the ability of wake steering controllers to optimize wind plant performance. In large farms, local conditions across the farm are needed to verify wake models and develop wake steering strategies to improve overall farm energy capture Barthelmie et al. (2009). Barthelmie et al. (2009) demonstrated the need for extensive pre-processing of data to establish the freestream flow, wind direction, and wind speed. Measurement biases also create challenges for accurate assessment of plant performance improvements Gaumond et al. (2014). While prior experiments typically use nearby reference turbines that are not misaligned (e.g. see Simley et al. (2021)), requiring reference turbines would limit which turbines at a wind plant can be incorporated into the wind farm flow control and the overall performance of the system.

<sup>&</sup>lt;sup>2</sup>Khoury College of Computer Sciences, Northeastern University, MA, USA

<sup>&</sup>lt;sup>3</sup>WindTester LLC., Portland, ME, USA & Roux Institute, Northeastern University, ME, USA

Preprint. Discussion started: 13 November 2025

© Author(s) 2025. CC BY 4.0 License.

Thus, a robust methodology is needed to experimentally determine the nacelle transfer function (NTF) to correct measurements from misaligned turbines from existing data sources Kanev (2020).

Existing NTFs in wind turbine controllers correct the measured wind speed and direction (often called static yaw misalignment correction) from sensors on the nacelle to estimate the upstream inflow wind conditions. These NTFs are typically determined experimentally for a turbine model using a nearby MET tower and are applicable for a turbine that is actively aligning itself with the wind. Because the sensors are mounted downstream of the rotor and in close proximity to the nacelle on the majority of upwind rotor turbines, the location of the sensor in the turbine wake and resulting measured wind speed and direction will be a function of the yaw angle of the rotor to the incoming flow. While prior research has addressed measurement errors from static misalignment Choi et al. (2018), flow distortion Rott et al. (2023), and dynamic yaw error Gaumond et al. (2014), the impact of intentional yaw offsets from wake steering on measured wind speed and direction remains largely unexplored. However, it is expected that intentional yaw offsets will affect both wind speed and relative wind direction measurements Cassamo (2022).

Developing reliable NTFs for wake steering requires accurate information about the inflow wind. Studies of wind turbine inflow condition measurements have incorporated multiple data sources including meteorological (MET) tower data Vad and Bottasso (2024) and LiDAR Fleming et al. (2014); Cortina et al. (2017). While these data sources can provide reliable upstream flow information, combining data from multiple sources introduces additional cost and complexity. In the context of wake steering applied at a commercial scale where wind farms may not contain MET masts near all turbines, it is beneficial to develop a data driven approach to estimate yawed turbine NTFs exclusively using wind turbine SCADA data Kanev (2020). In this study, we develop and evaluate a novel workflow to quantify sensor bias for wake-steered turbines relying only on turbine based sensors.

#### 45 2 Background

For turbines steered into the wind, NTFs are widely used to correct wind speed sensor biases due to the anemometer's position behind the rotor St Martin et al. (2017). These functions have been applied within the turbine controller to correct the measured wind speed downstream of the turbine to approximate the free upstream wind speed. The calibration of wind speed NTFs require an inflow reference data source (typically a MET mast). The use of a MET mast is part of the standard IEC 61400-12-2 approach for calibrating the anemometer of a wind turbine International Electrotechnical Commission (2012). As a result of several factors including temperature stratification and turbulence intensity, Martin et al. St Martin et al. (2017) found the need for condition-specific NTFs. Upwind data sources, rotor wake dynamics, and turbulent conditions make NTFs for wind speed difficult to calculate accurately, and thus NTFs are typically only an average across a range of conditions.

Similar challenges arise when calibrating the relative wind direction sensor. We will refer to this sensor measurement as a "wind vane" signal even though in practice both wind speed and direction are often measured using a sonic anemometer that has no moving parts. Existing wind vane corrections typically only focus on the  $0^{\circ}$  offset, also called the static yaw misalignment. Wind vane offset correction is similar to a wind speed NTF, in that both compare sensor measurements with free-stream wind

Preprint. Discussion started: 13 November 2025

© Author(s) 2025. CC BY 4.0 License.

65

conditions. While additional sensors including nacelle mounted LiDAR Simley et al. (2021); Fleming et al. (2014) may be used to measure static yaw misalignment, several methods been developed using just SCADA data. Réthoré et al. (2009) proposed a misalignment correction relying on the geographic placement of wind turbines. A recent approach to measuring systematic errors in wind direction measurements has been derived from yaw maneuver measurements Rott et al. (2023). Other methods use power curves to identify the yaw angle that maximizes power output Qu et al. (2022); Gao and Hong (2021); Pei et al. (2018); Karakasis et al. (2016). And some approaches identify yaw measurement outliers by analyzing wind direction and power distributions across turbines Astolfi et al. (2017, 2020a) or by utilizing anemometer data Astolfi et al. (2020b).

While wind direction and wind speed measurements are important with respect to their effect on wake steering experiments, more attention has been paid to investigating the power losses due to yaw misalignment and corresponding power gain for downwind turbines due to wake mitigation. For the upstream turbine, the power loss is greater under yaw offset operation for stable atmospheric conditions than for turbulent conditions Hulsman et al. (2022). FLORIS, the wake simulation tool from NREL, has been shown to predict the power loss due to wake interaction and potential uplift from wake steering comparably to results from field tests National Renewable Energy Laboratory (NREL) (2023); Simley et al. (2021). Simley et al. (2021) demonstrated that wake steering has a positive impact on farm energy production when the wind speed is low or when the atmospheric conditions are stable. Fleming et. al., Fleming et al. (2021) in an experiment with fixed yaw offsets, also demonstrated energy predictions in agreement with those predicted by the FLORIS model.

To optimize farm-wide power output, physics-based and data-driven flow control models predict optimal yaw strategies for wake steering Howland et al. (2022). Post et. al.Post et al. (2024) demonstrated that wake deflection under yaw angles of  $15-20^{\circ}$  shifts by approximately  $4^{\circ}$ , measured using a downstream turbine as a sensor. These experiments rely on limited SCADA channels and, when available, external meteorological observations Hansen et al. (2012). Sodar has been used to compare inflow with measured conditions at steered turbines Fleming et al. (2020). However, typically only SCADA data is available Réthoré et al. (2009), thus new methods for analyzing wake control data should not rely on met-masts or other sources of external reference data Cassamo (2022). For successful wake steering optimization in large wind farms, the uncertainty in wind direction must be taken into account Gaumond et al. (2014) and this uncertainty must be reduced as much as possible.

There is a need for wake-steering specific NTFs which use available SCADA data in order to asses the efficacy of wake steering and improve optimal yaw-strategies. Accurate measurement of wind direction and vane angles is critical for calibrating turbine specific NTFs. When available, advanced techniques may integrate multiple data sources, including SCADA and meteorological inputs Vad and Bottasso (2024). These approaches are particularly useful for accurate annual energy production (AEP) predictions and turbine performance assessments. However, when external instruments are unavailable, and detailed wind direction data is needed at the turbine level, iterative consensus methods based on SCADA data can be used to estimate the true wind direction for each turbine location Annoni et al. (2019). In developing a framework to assess the impacts of active wake steering, Kanev (2020) assumed that there would be no MET mast data source, emphasizing the need for data-driven methods. While the power gains from wake steering are critical, evaluating and analyzing sensor bias due to yaw misalignment remains a significant question for the application of wake steering. The NTFs for measuring relative wind angles (e.g., wind vane angles) have largely been applied to correcting static measurement errors rather than exploring sensor bias when

Preprint. Discussion started: 13 November 2025

© Author(s) 2025. CC BY 4.0 License.



yaw offset is intentional. Furthermore, estimating power output during collective wake steering remains a critical challenge, particularly for validating control algorithms under varying atmospheric conditions. In this study, we propose a novel NTF framework using only SCADA data from the turbines to predict non-yawed turbine measurements for key quantities of interest (QoI): wind direction, wind speed, and generator power.

### 3 Data and Methodology

Calculating reliable NTFs for wind direction, wind speed, and generator power during wake steering requires an estimate of the unsteered measurements to compare measured SCADA values against. Once those two data sources are identified, we compare them to create a function that maps the SCADA data gathered while steering into a predicted unsteered measurement. This study leverages data from a subset of a large windfarm where wake steering was implemented using a toggling experiment where steering was applied to some turbines for a portion of the time. First, we filter out data that might be biased for reasons other than wake steering. Second, we derive reference values for wind aligned turbine measurements of wind direction, wind speed, and generator power using SCADA data from neighboring turbines. Finally, we approximate the function which maps steered SCADA data to estimated unsteered measurements using machine learning for each of the measured QoI.

## 3.1 Wind Plant Description

SCADA and wake steering controller data recorded over 3.5 months at a large wind plant retrofitted with WindESCo Swarm<sup>TM</sup> hardware and software were used in this study. The layout of the turbines representing a portion of the wind plant is shown in Fig. 1. The predominant wind direction is from the SSW. Turbine identification numbers as indicated are arbitrarily assigned by WindESCo. Data from turbines 401, 403, 404, 406-412 were used for the analysis of steered turbine sensor measurements. These turbines are positioned on the front row of the farm with respect to the dominant wind direction. When wake steering is engaged every other hour on a toggle-switch program, these turbines were steered depending on the wind conditions. For southerly wind, they are not waked by other turbines in the farm, allowing for their data to be analyzed without the waking effects of upstream turbines affecting the sensor bias caused by the steering itself. Data was collected from January 1, 2024 to May 15, 2024 and included one-minute statistics from the turbines' SCADA system and the wake steering system. Aligning the two sets of data was accomplished through maximizing the correlation between the recorded wind speeds. The wake steering vane angle<sup>1</sup> was calculated by taking the circular difference between the wind direction and nacelle direction, both recorded by WindESCo Swarm<sup>TM</sup> hardware. A Global Navigation Satellite System (GNSS) compass on each turbine was incorporated by the wake steering system to calibrate the nacelle direction and wind direction measurements to true north.

<sup>&</sup>lt;sup>1</sup>While these turbines actually use sonic anemometers to measure both the wind speed and relative wind direction we will use the term "vane angle" in this paper as a more unambiguous description of the relative wind direction measurement which is also sometimes referred to as "yaw error" in the literature.

Figure 1. Layout for the southerly sector of the wind plant showing turbines. The predominant wind direction is  $\sim 180-210^{\circ}$  and only data with wind from this direction is used in the analysis. The steered turbines in the front row, 401, 403, 404, 406-412, experience the least wake effects from neighboring turbines.

### 3.2 Filtering Implausible Data





Removing data which contains bias sources other than wake steering ensures that estimating references values for the QoIs using neighboring turbines results in the most accurate estimates possible. After filtering to remove outliers and data from offline turbines, some data remained with incorrect measurements such as full-scaled values or erroneous direction measurements due to malfunctioning sensors. In order to remove implausible data, several calculated columns were added to the dataset. These calculated values for each Turbine of Interest (ToI) based on the data relative to that turbine and include (1) a reference wind direction value calculated from neighboring turbines, (2) the circular difference between the average measured wind direction and the reference wind direction, (3) the standard deviation of that difference, (4) a ratio of the ToI standard deviation over the standard deviation of calculated reference value, and (5) a score defined by the number of standard deviations of the ToI wind direction from the mean wind direction of the entire farm at each timestamp in the data. Filter columns were calculated using a rolling window (except for the outlier weight column) and have the same frequency (one minute averages) as the original data. The averages were calculated using circular arithmetic for the wind direction. The reference value for each QoI is derived by taking the average of the nearest 25 measurements at the same timestamp with the ToI excluded. We computed a rolling average with a window of 8 minutes before applying the difference and standard deviation functions to the data. Meanwhile, the outlier weight is calculated for each timestamp without a rolling average.

Each filter column targets a different type of implausible data. We include the rolling mean difference of the QoI to remove measurements that have a sustained discrepancy from their neighbors. We include the rolling standard deviation of the difference in order to quantify the variability between the reference value and the measured value, which could indicate a



**Table 1.** Thresholds for Filter Columns

| Column                                                  | Threshold | Type  |
|---------------------------------------------------------|-----------|-------|
| Wind Direction Rolling Mean Difference                  | top 10%   | upper |
| Rolling Standard Deviation of Wind Direction Difference | top 10%   | upper |
| Ratio of Standard Deviations of Wind Directions         | top 10%   | upper |
| Outlier Score for Wind Direction Variability            | top 10%   | upper |
| Generator Power                                         | 200 kW    | lower |
| Wind Speed                                              | 2.5 m/s   | lower |
| Wind Direction                                          | 150 deg   | lower |
| Wind Direction                                          | 240 deg   | upper |

measurement problem. The ratio of ToI standard deviation to the reference's standard deviation highlights scenarios when the ToI measurement is experiencing unusually high variability relative to its neighbors, which could signal a sensor issue. Finally, the outlier score gives a measure of how unusual a measurement is at a particular time, assuming measurements at each time follow a normal distribution. In addition to these calculated columns, we also filter on existing values in the data, such as the generator power and the wind speed to only include operating conditions of interest. When the generator power is extremely low, it indicates that a turbine is not functioning in normal operating conditions. Also, at very low wind speeds, the wind direction and wind speed across the farm can be more variable; thus, it is likely that the estimation of a reference value might be less accurate. The filters are applied with thresholds shown in Table 1 to remove most implausible data and data that is not of interest for the subsequent NTF analysis. For the first four calculations listed in Table 1, the selected threshold excludes the highest 10% of the data. The thresholds overlap, so remaining data is in the intersection of all eight criteria.

## 3.3 Estimating Non-misaligned Data Values

Four weighted average algorithms shown in Table 2 were evaluated to estimate the current wind speed, wind direction and power at the ToI based on the data from neighboring turbines. The predictive capability of these methods are evaluated using data gathered in non-wake steering operations. The best performing algorithm is then selected to estimate the unsteered wind speed, wind direction, and power during steering events. Our dataset includes measurements recorded every minute, but due to filters and data anomalies data from every turbine is not available for every minute. For a given ToI, data points where fewer than 10 nearby turbines have usable data are not included. Estimation was conducted independently for each turbine and each timestamp, leaving out the ToI in the calculations as well as any steered neighboring turbines. A turbine is considered steered if the wake steering control target vane angle magnitude exceeded 0.1 degrees.

The data from neighboring turbines are ordered by their distance to the ToI, and in general, more distant points are assigned less weight and are less important to the predicted QoI. The relative location of each turbine within the farm is converted from meters into rotor diameters, and the distance between the ToI and all other turbines is calculated. In Table 2, N is the number

© Author(s) 2025. CC BY 4.0 License.



**Table 2.** Estimation Methods for unyawed signals at ToI based on neighboring turbine data  $(i \in [1:N]$  excludes ToI)

| Method                                    | Equation                                                                                                                        | Weights                                                                                                           | Parameters |
|-------------------------------------------|---------------------------------------------------------------------------------------------------------------------------------|-------------------------------------------------------------------------------------------------------------------|------------|
| Global Average                            | $\frac{1}{N} \sum_{i=1}^{N} x_i$                                                                                                | NA                                                                                                                | NA         |
| Gaussian Weighted Average                 | $\frac{\sum_{i=1}^N w_i x_i}{(\sum_{i=1}^N w_i)}$                                                                               | $w_i = \frac{1}{s\sqrt{2\pi}} \exp\left(-\frac{1}{2} \left(\frac{ \mathbf{y_i} - \mathbf{y} }{s}\right)^2\right)$ | s = 18     |
| Inverse Distance Weighted Average         | $\frac{\sum_{i=1}^{N} w_{i} x_{i}}{(\sum_{i=1}^{N} w_{i})}$ $\frac{\sum_{i=1}^{N^{*}} w_{i} x_{i}}{(\sum_{i=1}^{N^{*}} w_{i})}$ | $w_i = \frac{1}{ \mathbf{y_i} - \mathbf{y} }$                                                                     | NA         |
| Cluster Inverse Distance Weighted Average | $\frac{\sum_{i=1}^{N^*} w_i x_i}{(\sum_{i=1}^{N^*} w_i)}$                                                                       | $w_i = \frac{1}{ \mathbf{y_i} - \mathbf{y} }$                                                                     | $N^* = 8$  |

of turbines in the farm,  $y_i$  is the location vector in rotor diameters of neighboring turbine i with respect to the ToI, and y is the location of the ToI. For cluster averaging we chose the eight nearest turbines, N\*=8.

## 3.4 Modeling the Nacelle Transfer Function

The objective of each NTF is to take sensor measurements for a QoI as input and produce the corresponding value as if the turbine were not steered at the time. In order to fit the NTF, the predicted reference values are determined by neighboring-turbine estimation and compared to the measured values. An underlying assumption in this analysis is that the NTF is a function of the wind turbine model geometry and operation and therefore, in constructing a NTF, we combine the data from multiple turbines of the same model. However, in order to ensure that static bias present in each turbine (as detected through the estimated QoIs) does not influence the NTFs, we undertake the following procedure to combine estimated data across different turbines. First, the estimated QoI from the case where the ToI is not steered is gathered. For each estimation method tried, the data is fit to a linear model, with the independent variable being the original sensor value and the dependent variable being the estimated value of the sensor. The y offset of this model is then subtracted from the estimated data for both the unsteered and steered data. Then, when the data are combined from multiple turbines, the estimated static bias is removed.

#### 3.4.1 Bagged Tree Regression

In order to explore the effects of multiple variables including turbulence intensity and to enable non-linear fitting of the NTFs for the three QoIs, we chose to model the NTF using a machine learning algorithm, specifically a bagged tree regressor. In this algorithm, a set of binary trees are constructed using the training data. Each tree recursively splits its sample of the training data as each node is 'grown', attempting to minimize the difference of the predicted output in the samples grouped into each of its 'leaves'. After training the trees, the forest makes a prediction by taking the mean of individual tree predictions for a given instance of data Breiman (2001). Bagging, or bootstrap aggregation, is a method that lessens the tendency of decision trees to overfit their dataset through independent samples of the training data provided to each tree in the ensemble. In addition to using the random forest for regression, an additional benefit of the machine learning model is its use as an effective selector of predictors, or input variables, as demonstrated by Genuer et. al. Genuer et al. (2010). A deeper background on RFs is covered

© Author(s) 2025. CC BY 4.0 License.





**Table 3.** Predictor Variables for the Nacelle Transfer Functions

| Target      | Estimated Wind Vane  | Estimated Wind Speed | Estimated Power      |
|-------------|----------------------|----------------------|----------------------|
| Predictor 1 | Measured Power       | Measured Wind Speed  | Measured Power       |
| Predictor 2 | Turbulence Intensity | Turbulence Intensity | Turbulence Intensity |
| Predictor 3 | Measured Wind Vane   | Measured Wind Vane   | Measured Wind Vane   |

in the work by Breiman Breiman (2001). For the code used in this paper to implement the regression model, we used the BaggingRegressor function from the Python library scikit-learn Pedregosa et al. (2011).

## 3.4.2 Machine Learning Modeling Set Up

A three step process was implemented to set up and validate the NTF machine learning model for each QoI (wind speed, wind vane, and power). The predictor features (parameters) are selected for each model as shown in Table 3. Because wind speed and power are highly correlated due to the operation of the turbine controller, only one of the two is included at a time as a feature in each model. We use measured wind speed to predict the estimated true wind speed and measured power to predict the unyawed power. For wind vane, either wind speed or power could be selected, and power was chosen. The best weighted averaging method is used to predict the true value during wake steering, and these "estimated" values are used for the machine learning targets: estimated true wind vane, estimated true wind speed, and estimated true generator power. For each QoI, a 5-fold cross validation was used to first tune the model hyper-parameters and then to estimate the uncertainty of the resulting predictions. A final model using the same hyper-parameters was then trained using all the data to provide the best possible curve fit.

The hyper-parameters include the number of base estimators, the number of samples to train each base estimator, and the number of features to train each base estimator. The data was sorted by timestamp in chronological order and then split into 5-folds chronologically in time. This approach avoids having artificially similar train and test data due to the high level of autocorrelation in wind turbine time series data resulting in higher confidence in the model uncertainty. Hyper-parameter tuning was accomplished with <code>GridSearchCV</code> from scikit-learn. Across the folds for each parameter combination, we calculated the average test root mean square error (RMSE) and selected the parameters associated with the lowest average RMSE.

After identifying the optimal hyper-parameters, the k-fold test predictions and data for these parameters were collected together and used to evaluate the prediction accuracy. In addition to calculating the average RMSE error metric, we also calculated the correlation coefficient between the predicted and true values. A final NTF model was then trained for each QoI using all of the data to provide the best fit of the available data.

**Figure 2.** The distribution of the standard deviation of the rolling circular difference and circular mean difference for wind direction using data from turbines 408-413. Most of the data points are concentrated near the origin. This reflects that in the majority of the data set, similar wind direction measurements prevail for groups of neighboring turbines. There are clouds of data at the extreme circular differences that have correlated high circular standard deviations. These data points are assumed to have unreliable sensor measurements and are excluded from further analysis.

### 4 Results


## 4.1 Filtering

Focusing on good-quality data from turbines operating normally was critical for testing our estimation methods. Figure 2 shows the distribution of the filtering columns before truncating the data. When the mean difference is close to zero, the standard deviation of that mean difference is also low. This implies that when wind conditions are variable across the farm, the discrepancy between nearby turbine measurements will be more variable. Using this insight, we based our estimation process on data from stable conditions.

© Author(s) 2025. CC BY 4.0 License.

Table 4. Comparing Dataset Size Before and After Filtering

| QoI                    | Number B | Before Filtering | Number A | After Filtering | Percent I | Remaining |
|------------------------|----------|------------------|----------|-----------------|-----------|-----------|
|                        | Steered  | Unsteered        | Steered  | Unsteered       | Steered   | Unsteered |
| Wind Direction         | 99,069   | 9,961,474        | 19,913   | 2,013,142       | 20.10%    | 20.21%    |
| Wind Speed             | 97,664   | 9,904,272        | 19,893   | 2,010,497       | 20.37%    | 20.30%    |
| <b>Generator Power</b> | 98,114   | 9,939,542        | 19,904   | 2,012,887       | 20.29%    | 20.25%    |

After generating the filter columns and applying the thresholds described above, we divided the data into steered and unsteered categories. Table 4 lists the total number of data points from the ToIs for the steered and unsteered data partitions before and after filtering on each QoI. Turbulence intensity was estimated as a global average of the 10-minute wind speed standard deviation divided by the 10-minute average wind speed and thus was available for every data point.

Based on the thresholds set in Section 3.2, a significant percentage of data was removed in order to focus on testing the estimation procedure in ideal conditions where the flow across the farm was steady. To highlight the properties of the data, we plot the filter columns over time, as seen in Fig. 3. In the first row of Fig. 3, the circular difference from the average for all three turbines is consistently close to zero with a few short-term departures. This indicates that the wind vanes and nacelle positions combined are measuring a similar wind direction to their neighbors. The spikes in the circular standard deviation reveal measurements which are changing rapidly with respect to the reference direction. Turbulent periods coincide with high circular differences. The ratio of the standard deviations is more constant than the standard deviation of the circular difference. This is because in turbulent wind, the fluctuating wind direction is experienced by the ToI and the neighboring turbines alike, resulting in a ratio between the two quantities closer to one. The last plot shows that the turbines have measurements within one standard deviation for most of the time selected. This last column highlights moments where the wind direction measurement for a given turbine is unreliable.

#### 4.2 Estimation





Under normal operation, turbines maximize their own power by aligning with the wind. We compare the estimation methods on unsteered data because our goal is to use measured QoIs during wake steering to predict what the turbine would have recorded if it were aligned with the wind. The method that is most accurate at estimating a reference value when the ToI is unsteered can be taken as the method to be used on the steered partition of the dataset, allowing us to build the NTF using this data. In Table 5, we show the different consensus methods applied to the different QoIs on unsteered data. The Gaussian Weighted Average method performed the best for the wind direction estimation and wind speed, while the Cluster Inverse Distance Weighted Average method performed the best for the power. Note that all of these methods have some degree of error, but it is within a reasonable scale compared to the variability of measurements for each QoI. We see that for each QoI, the farm-wide or global average is one of the least accurate methods for estimating the reference value.

**Figure 3.** Example plots of filter calculations associated with the wind direction for wind turbines 401, 416, and 456 for a selected 18-hour time period. From top to bottom the plots show the circular mean difference, the standard deviation of the circular mean difference, the ratio of the standard deviation of the ToI signal over the reference signal, and the standard deviations from the mean.

Table 5. Comparing consensus method results for Turbine 401

| QoI (unsteered)                  | Wind Direction CRMSE | Wind Speed RMSE | Generator Power RMSE |
|----------------------------------|----------------------|-----------------|----------------------|
| Global Average                   | 5.86                 | 1.52            | 343.37               |
| Gaussian Weighted                | 5.32                 | 1.33            | 298.59               |
| Inverse Distance                 | 5.45                 | 1.39            | 314.13               |
| Inverse Distance with clustering | 5.40                 | 1.39            | 269.33               |

Figure 4 plots the combined data from turbines 401, 403, 404, 406-412 with the measured QoI value on the x-axis and the estimated QoI value on the y-axis. While difficult to read due to the sheer number of data points, the densest area of data for



**Figure 4.** Left to Right: scatter plots of estimated vs. measured wind vane, estimated vs. measured wind speed, and estimated vs. measured power using unsteered data for selected turbines. Estimation methods were Gaussian Weighted Average, Gaussian Weighted Average, and Cluster Inverse Distance Weighted Average, respectively.

all three QoIs is around the line y=x, which points to the reliability of the estimation method. When looking for bias in wind direction measurements in Section 4.3, we compared the measured and the estimated wind vane instead because the wind vane sensor bias is caused by the difference between wind direction and nacelle direction, changing the flow of air around the rotor and the area of the nacelle directly behind the rotor Rott et al. (2023). There is still considerable noise in all QoIs, especially for wind vane, which has a wide area of uncertainty around the origin. For the wind vane, most of the data centers around zero, which is expected for unsteered data. For wind speed, most of the data appears to be at the range of approximately 7.5 m/s. At the higher end of the wind speed range, the data appears more irregular, with a more significant portion of the estimated wind speed greater than the measured wind speed.

A kernel density estimate (KDE) plot is a graphical representation of the probability density of a continuous variable approximated by summing kernel functions. Figure 5 uses two-dimensional KDE plots to provide deeper insight into the estimation results. On the left, we compare the measured to estimated QoIs for unsteered data, and on the right, the same comparison is made for the steered data. Both plots look similar, and the density around the diagonal continues to support the use of the estimation methods. Row three shows the generator power KDE plots, but most of the data is at the rated power level, making it difficult to determine any trends. By filtering the power level in row four, more distinguishing features are present. Notice that the steered data in row four has an area of density around 1750 kWs where the estimated power is greater than the measured, which could be a direct result of wake steering reducing the power generated by the steered turbine.

#### 4.3 Nacelle Transfer Function Modeling

We use machine learning to investigate the relationship between measured and estimated values of the QoIs. The objective is to compare measured wind direction, wind speed, and generator power to an estimate of the same variable where the turbine is not steered to see how the SCADA data is changed by yawing. The estimation methods described in Section 3.3 have been shown to predict accurate trends on unsteered data where we can verify the estimated QoI with the known measured QoI.

**Figure 5.** Top to bottom: KDE plots of the estimated vs. measured wind vane, estimated vs. measured wind speed and estimated vs. measured power for both the unsteered (left) and steered (right) data using the Gaussian Weighted average for wind direction and wind speed estimation and inverse distance weighted average with clustering for generator power. The last row of figures shows the estimated vs. measured power filtering out measured power above 2200 kW, which helps to better visualize power behavior below the turbine rated power.

Preprint. Discussion started: 13 November 2025

© Author(s) 2025. CC BY 4.0 License.






Taking the best-performing estimation methods, we combined the data from multiple turbines together as described in Section 3.4 to estimate the unsteered QoI for each steered measurement in the dataset.

We examined statistical relationships using KDE plots for both steered and unsteered data, as shown in Fig. 6, to assess the relationships inferred by the NTF fitting procedure. In all cases, the steered data plots were derived from a smaller amount of data than the unsteered plots. Column one, rows one and two depict the ratio of estimated to measured wind speed as a function of wind speed. The measured values appear to take on values roughly 5 m/s and higher; this is a result of filtering out lower wind speeds. The ratio of estimated to measured values appears to show that the estimated values are slight underestimations of the measured values on average. These two observations are evident in both the unsteered and steered data, with the latter observation being more pronounced in the steered data. In the unsteered data, there are relatively more data at higher wind speeds when compared to the steered data.

Looking at the column two rows one and two, the ratio for power as a function of measure power, inclusive of all power values, illustrates that the measured values appear to take on values predominantly around 2500 kW, and the ratio of estimated to measured values situates predominantly around 1.0; these observations hold true for both the unsteered and steered data. Column three rows one and two show the plots for the power ratio excluding data with power values 2200 kW and greater. In these plots, the density around measured values of power is more uniform in the range from roughly 400 kW to 2200 kW. The ratio of estimated to measured values also situates predominantly around 1.0 for both steered and unsteered data. The steered data shows that the estimated values are slight overestimations of the measured values. Also, the steered data appears to have a greater density of data around the power level of 1200 kWs compared to the unsteered data. This is likely due to the behavior of the wake steering algorithm and the conditions for when it activated. One thing to note with both the wind speed and power plots is that, while the unsteered data exhibits more scattered high-density data, the steered data exhibits high-density data in more concentrated areas (e.g. 10-12 m/s for wind speed, 1200-1400 kW for power excluding values 2200 kW and higher), which aligns with expectations.

Rows three and four include plots illustrating measured wind vane vs. a measured QoI. Column one rows three and four show the distribution of data for wind speed vs. wind vane. Both the unsteered and steered data are dense about a measured wind vane of 0 degrees, with some concentrations around slightly negative measured wind vane values, particularly for the unsteered data. This might be a sign of misalignment. In the steered data, the area of density is more spread out across the range of wind vane values, which could be an effect from steering to redirect the wakes downstream.

Looking at column two rows three and four, the plots for power show a concentration of high-density data from a measured wind vane range of roughly 2 to -5 degrees, which is true for both the unsteered and steered data. Column three rows three and four show the same information excluding power values 2200 kW and higher. In these plots, the concentration of high-density data appears in a measured wind vane range of roughly 5 to -5 degrees, which is true for both the unsteered and steered data. One additional thing to note with both the wind speed and power plots is that the measured wind vane values for the steered data exhibit wider ranges than the values for the unsteered data, which aligns with expectations.

Figure 7 shows the feature importance chart for each NTF. Each model relies primarily on one feature, the measured QoI value. The NTF for the wind vane has the most contributions from the other predictors among the three NTFs. This supports

Figure 6. KDE plots for QoI relationships. First row: column one shows measured wind speed vs. the ratio of estimated to measured wind speed for unsteered data, column two shows measured power to the ratio of estimated to measured power for unsteered data, column three shows the same as column two but for power measured below 2200 kWs. Second row: same quantities measured as in row one but for steered data. Third row: column one shows measured wind vane vs. measured wind speed for unsteered data, column two shows measured wind vane vs. measured power for unsteered data, column three shows the same as column two for power measured below 2200 kWs. Fourth row: same quantities measured as in row three but for steered data.

**Figure 7.** Relative importance of each predictor via the random forest models used to create the NTFs. For wind vane prediction, the measured wind vane is the most important input, but there are contributions from both measured power and turbulence intensity. For wind speed, the measured wind speed is the most important. Finally, for power, the measured power is the most important input factor.

Table 6. Results from Fitting NTFs



|                     | Predicted wind vane angle | Predicted wind speed | Predicted power |
|---------------------|---------------------------|----------------------|-----------------|
| <b>Testing RMSE</b> | 4.0792                    | 1.0950               | 211.3274        |
| Testing $R^2$       | 0.8345                    | 0.9595               | 0.9584          |

the existence of nonlinear interactions between wind speed, turbulence intensity, and the measured wind vane due to flow distortion from the rotor and nacelle.

As described in Section 3.4.2, the predictive model was validated using k-fold cross-validation and the resulting overall RMSE and correlation for the combined k-fold test cases are shown in Table 6. Residual plots shown in Fig. 8 show a significant number of points with large prediction errors. However, the correlation coefficient of the testing data is high, and the density plots of the residual in Fig. 8 explain why the picture is misleading; many of the points with accurate predictions are hidden in the data cloud.

In Fig. 9, we show the resulting NTF for wind vane plotted for selected values of turbulence intensity and power while varying the measured wind vane angle from -20 to 20 degrees. The NTF for the wind vane in the first row shows a clear bias for positive measured wind vane values. Compared to the function y = x, which would represent an exact match between the estimated wind direction and the measured wind direction, the function that the model predicts for these turbines reveals that the wind vane overestimates the measured wind direction for positive wind vane angles. The measured wind direction is more accurate for negative vane angles up to about -15 deg. After this the model shows significant deviation but that may be due to a lack of data to provide accurate training of the machine learning model in this region. The observed trends are similar for different power generation levels, but more pronounced for the higher level of generated power. The asymmetry in the observed wind vane NTF could be due to the asymmetry in the mounting location of the sonic anemometer on the nacelle as well as flow asymmetry created by the rotor rotation direction.

In Fig. 10, we show the NTF for wind speed. For this and the NTF for power, we took a different approach, plotting the ratio of the measured over the predicted QoI as a function of the wind vane. For wind vanes greater than zero and higher wind

**Figure 8.** Residual plots across folds for the three NTF models as a way to gauge the uncertainty of the plots in Figure 9, 10, and 11. The residual mean and standard deviation for the plots for wind vane, wind speed, and power are -0.153 and 4.08, 0.018 and 1.10, and 8.96 and 214, respectively, keeping in mind that each variable has not been scaled. The wind direction estimation is the most uncertain relative to the scale of its measurements.


**Figure 9.** Inference plot of the NTF for wind vane measurements at four different power levels showing the measured value on the x-axis and the predicted unsteered value on the y-axis.

speeds, the measured wind speed is greater than the predicted, while for wind vanes less than 0, the opposite relationship holds for all wind speeds, except at very negative wind vane angles. The model predicts undermeasurement of the wind speed for all wind vane angles at 7 m/s which is unexpected and suggests some uncertainty in these results since all of the plotted curves would be expected to go through [0, 1]. One possible explanation is that the filtering has biased the data set toward points where the neighboring turbines measure higher wind speed than the ToI at speeds near the lower bound of data used.

In Fig. 11, we show the NTF for power. For the power predictions, except for measured power at 2000 kW, as the absolute value of the wind vane increases, the ratio decreases, meaning that the measured power is less than the predicted. This makes sense because during yawing, the turbine generates less power, especially at lower power levels. Notice that this relationship is symmetric except for measured power at 2000 kW although again at lower power levels the prediction appears to be biased to over predict the power of the ToI. Again this bias may be due to how the data was filtered because all turbines had to be generating power in order to accept a data point.

**Figure 10.** Inference plot of the NTF for wind speed showing the NTF estimated over measured wind speed ratio, as a function of measured wind vane angle for four different measured wind speed values.

**Figure 11.** Inference plot of the NTF for power showing the NTF estimated over measured power ratio, as a function of measured wind vane angle for four different measured power values.

https://doi.org/10.5194/wes-2025-223
Preprint. Discussion started: 13 November 2025

## 5 Discussion







The three major components of this study - filtering, estimation, and NTF modeling - contribute to the understanding of bias in sensor data as a result of intentional yaw misalignment. Our results highlight both promising trends and challenges that warrant further investigation in order to analyze the bias from yaw misalignment and its implications for wake steering.

Noise from various sources makes it challenging to use neighboring turbine SCADA data to estimate measurements at a specific ToI. Stringent thresholds for removing implausible points were imposed to address this issue. However, a result of this filtering process is that the remaining data might not be fully representative of the behavior under all conditions and may bias the dataset toward more stable conditions and higher wind speeds and power levels. The methods described in Section 3.2 focused on identifying major discrepancies between a turbine and its neighbors. The benefit of this approach is that it does not require an external inflow measurement, instead, relying solely on data from the farm itself. The filtering method was successful in removing data points where the ToI deviated significantly from the average of its neighbors for a particular QoI value. Regarding the standard deviation of the difference or the ratio of standard deviations, these factors excluded data points from periods of high volatility. This reinforces the analysis by retaining data with steadier trends, which improves the performance of the estimation methods. In the future, filtering features could be shared across the different QoIs, such as using a feature concerning generator power to filter wind vane angle data. Filtering could be further improved by applying machine learning to automatically derive thresholds based on outlier detection. Techniques such as change-point detection Aminikhanghahi and Cook (2017) could also be employed for more reliable filtering.

Our approach to estimate local wind conditions requires only neighboring turbine data, which is crucial when an external data source, such as a MET tower, is unavailable. We explored several estimation methods, and the filtered, unsteered data provided evidence that these methods were effective. For each QoI, we selected the best-performing method and applied it to the steered data. It is worth noting that different choices of filtering thresholds might have resulted in different rankings of the estimation methods in Table 5. The most effective methods were the Gaussian Weighted Average method for wind direction and wind speed, and the Cluster Inverse Distance Weighted Average method for generator power. These methods excelled because the conditions at the nearest turbines were most similar to those at the ToI. All of the proposed methods outperformed the global average for each QoI. The effectiveness of these simple methods is encouraging, as they can be calculated extremely quickly. However, more work is needed to enhance their estimation accuracy, and additional techniques that rely on other machine learning models or the consensus algorithm developed by Annoni et al. (2019) should be explored.

In Fig. 4, the data cloud for estimated power vs. measured power has an unusual rectangular shape. This occurs because the rated power of the turbine generator limits the performance to this level as the wind speed increases. If the ToI does not produce power at the rated level, then the y-coordinate of the point will be at 2500 kW, but the x-coordinate will be below this. This results in a horizontal cluster around y = rated power. Similarly, the cluster around x = rated power can be attributed to underestimation. That is, the consensus is lower than the rated power while the QoI is the same. Despite these challenges, for the unsteered data, reasonable estimates of measured SCADA values for wind direction, wind speed, and generator power were achieved (see Table 6), allowing the application of this method to the steered data. For wind direction, we were able to predict

Preprint. Discussion started: 13 November 2025

© Author(s) 2025. CC BY 4.0 License.






the measured unsteered values within approximately 5 to 6 degrees. Applying this method to cases where the ToI was steering, we would expect a similar level of error in the estimation for any given data point. However, the bias in these estimates should be much smaller in aggregate for the trained machine learning model.

We used machine learning to approximate the NTF for the QoIs and to compare the discrepancy, in aggregate, between measured and reference values across the set of studied turbines. The underlying assumption is that the NTF is the same for any given model of turbine, and not specific to the individual turbines. These results must be interpreted in the light of the filtering and estimation process. The data used to train the NTF models comes from only a relatively small number of turbines operating for a short window of time. Thus, the observed relationships must be interpreted as exploratory rather than definitive. Furthermore, the filtering process reveals that choosing different thresholds for accepting or rejecting data has an effect on the estimation process for calculating reference values. There are many directions to pursue with regards to improving the estimation process. With a more robust estimation process for reference values, a greater percent of the steered data could be included in training the NTF. With these caveats, evaluating the results of the estimation process still reveals interesting relationships. Using the bagging regression model to analyze the data, we observed bias in the measured vane angle and wind speed. Specifically, for turbines that are intentionally yawed away from the wind direction at extreme steering angles of  $\pm 15^{\circ}$ , the model predicts discrepancies between measured and estimated wind direction. At  $+15^{\circ}$ , the NTF predicts an over measurement of relative wind direction, while at  $-15^{\circ}$ , it shows a more accurate measurement. At angles  $

for increasing negative yaw errors, which is also seen in the plot for wind speed nacelle measurement error in Fig. 2 of Kanev (2020). Our NTF model suggests a potential bias of 2.5% to 9% in wind speed measurements compared to the unsteered measurements of the nearest turbines. Similarly, Kanev (2020) found for  $-20^{\circ}$  yaw offset the wind speed measured at the yawed nacelle was smaller than the free-stream wind speed measured by the MET mast. For measured windvane of  $-20^{\circ}$  and wind speeds less than 16 m/s (where the training data was denser), our model predicts a ratio below one for measured yawed wind speed over unsteered wind speed. The unsteered wind speed measurement is larger than the measured yawed wind speed, aligning with their findings where the unyawed measurement for wind speed was greater than the yawed measurement Kanev (2020). We also observe that the NTF predicts different biases for different measured wind speeds, which demonstrates the relationship between yaw angle and wind speed combines to influence potential sensor bias.

Meanwhile, Fig. 11 shows the NTF for power at 500 kW predicts power values consistently larger than the measured ones, which might stem from insufficient data in the training set. At 2000 kW, the measured power for negative wind vane angles exceeds the predicted values, deviating from the trends seen in other curves. This indicates differing biases at various power levels. At power levels of 1500 kW and 1000 kW, the trends are similar: as the wind vane angle increases or decreases, the predicted power is greater than the measured power, with a smaller ratio observed at the lower power level. This aligns with expectations, as it demonstrates that during wake steering, the measured power is lower than what would be expected if the turbine is aligned with the wind.

#### 6 Conclusions






Accurately characterizing sensor bias under steered conditions is crucial for improving future wind farm flow control algorithms and refining the analysis of wake steering performance. It is desirable to derive the NTFs for correcting this sensor bias from SCADA data alone, particular for turbines and wind farms that are retrofit to enable wake steering where additional sensors such as MET towers are not available. To address this, we demonstrated a workflow to empirically filter and estimate reference wind condition measurements and then derive ML based NTFs comparing these estimated values with the measurements on steered turbines.

Results show significant bias is present in the measurements of relative wind direction and wind speed on intentionally yawed turbines. In the case of the wind vane signal, the vane angle is over reported for positive yaw angles while it is more accurate for negative vane angles between -10 and -15 degrees for this particular turbine model. Higher turbine power only created a small increase in the magnitude of the bias. Wind speed measurements are observed to be biased by 5 to 10% as a function of the vane angle and wind speed, and the wind speed is generally over-reported for positive vane angles and underreported for negative angles. Power is reduced with increasing vane angle by 10 to 20% but is observed to be non-symmetric with greater losses for positive vane angles than for negative vane angles and the loss is smaller for higher power levels in this data set. In particular, little power loss is observed between +5 and -10 degrees. This behavior is not well represented in existing estimates used in wake optimization software such as FLORIS and an empirically generated NTF for power may be beneficial in improving flow control optimization in the future.

Preprint. Discussion started: 13 November 2025

© Author(s) 2025. CC BY 4.0 License.



Several discrepancies in the resulting predicted NTFs are noted including bias at 0 vane angle for both wind speed and power at lower wind speed and power levels. This may be due to filtering creating bias in the training data set due to removing turbines that are not generating any power. Trends for large negative vane angles beyond  $-15^{\circ}$  also appear inconsistent and may be due to a lack of sufficient training data in that part of the feature space.

Further work is needed to validate the robustness of these approaches and improve the data filtering. Potential directions include, exploring change-point detection or automatic outlier detection to select data for inclusion without hard limits that can bias the results. Investigating processes for efficiently monitoring changes in variance and covariance in the time series data of groups of turbines would be a promising direction. Further evaluation of different consensus methods for estimating reference conditions is another promising direction to expand on the current study. Finally additional exploration of the results in terms of the feature space and characterization of the uncertainties of the NTF as a function of those features is necessary to ensure only reliable results are used in incorporating these approaches into a larger analysis or control project.

Code and data availability. The recorded SCADA and wake steering controller data set used in this study is proprietary to WindESCo and can not be made available. Post-processed data for NTF analysis and plotting along with the full analysis code and plotting tools are available at this link Abbatessa et al. (2025)

Author contributions. Algorithm development, data filtering, post-processing, and writing by A. Gettemy. Additional analysis and plotting by L. Abbatessa. Guidance, Advising, and editing by N. L. Post

Competing interests. The authors declare that they have no potential conflict of interest.

Acknowledgements. WindESCo Inc. kindly provided the experimental data set from a wind farm where WindESCo Swarm<sup>TM</sup> WFFC was implemented. Funding for this research was provided by Northeastern University Roux Institute and the Herold Alfond Foundation.

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
