# Peer review of "Evaluating Yawed Turbine Transfer Functions from SCADA Data"

_Wind Energy Science, 2025_

## Referee Comment (RC2)

This manuscript presents a SCADA-based approach for estimating nacelle transfer functions (NTFs) describing the effect of intentional yaw misalignment on measured wind direction, wind speed, and power. The authors first test several consensus-type estimators that use neighboring turbines to infer yaw-aligned operating conditions and then use the selected estimator to generate reference values for training a machine learning model. The ML model maps measurements from yaw-misaligned turbines to the inferred yaw-aligned quantities.

The topic is relevant, particularly in the context of increasing use of wake steering and the lack of external inflow measurements at many wind farms. The dataset is substantial and the general idea is interesting. However, there are several issues with the methodology that currently limit confidence in the results. Additionally, the lack of benchmarking against existing methods for modeling yaw-misaligned wind turbines makes it difficult to assess the contribution of this work. The manuscript is also difficult to follow due to unclear terminology and insufficient explanation of methods and results. Significant improvements in methodology, benchmarking, and clarity are necessary before the manuscript can be considered for publication. I would be happy to review the manuscript again once these issues have been addressed.

**Major Comments**
1. The proposed ML-based NTFs are not compared against any established models for yaw-misaligned wind turbines. There is no comparison to simple cosine law relationships which are the state-of-practice method (Gebraad *et al.* 2016). Additionally, there is no benchmarking against conventional NTF calibration methods based on met towers or LiDAR. Since the stated goal is to replace met-tower-based calibration, the absence of any validation against a met-tower-derived NTF significantly weakens the contribution. Without such benchmarks, it is difficult to judge whether the proposed model is useful in place of existing methods. These existing methods should be discussed in the introduction and at minimum a cosine model should be used for a baseline comparison.
2. A key assumption in this work is that yaw-aligned turbine quantities can be reconstructed reliably from neighboring turbines at one-minute resolution. Temporal and spatial variability can lead to differences between turbines that are unrelated to yaw misalignment. In turbulent flow, averaging may reduce noise more effectively than aggressive filtering. A sensitivity study to time-averaging the data using various averaging windows (e.g. 2 min, 5 min, 10 min, etc.) could be performed.
3. The final NTF model is trained using the full dataset (L194, L204) and as a result the model is not tested on any unseen data. The best practice for ML model development is to use separate datasets for training, validation and testing. Since the model is not tested against a holdout test dataset it is impossible to know if the model has been overfit to the data and therefore how it will perform on unseen data. As a result "testing RMSE" in Table 6 is misleading since the model has been trained on the test data. This is a major oversight in the ML methodology of the work. The model should be trained,

tuned, and tested on three separate subsets of the wind farm data and the model should never receive information about the testing dataset before the final evaluation.

4. The parameters of the weighted averaging methods (e.g., Gaussian width, number of clustered turbines) appear *ad hoc* (Table 2). A sensitivity analysis could be helpful to justify these choices.

5. Figures 5 and 6 contain many subfigures making it difficult to digest the information or understand what the intended key takeaways are from the plots. Please consider reducing the number of subplots and more explicitly stating what trends the reader should notice.

6. Figures 10 and 11 show that the wind speed and power ratio of the model does not pass through zero for some cases as would be expected. Additionally, since the model is not tested on an unseen holdout test dataset it is not possible to tell if the NTFs represent meaningful and repeatable empirical relationships or overfitting to the data.

**Minor Comments**

- Section 3.2 would benefit from explicit equations defining the filtering metrics. In general, the methods section is text-heavy and could benefit from more equations which concisely describe the methods introduced.

- Figures 2 and 4 would be clearer if point density were shown directly (e.g. using a KDE plot).

- The terms "predicted" and "estimated" are ambiguous since there are two models in use: the neighbor-based yaw-aligned quantity estimator and the NTF model. It may be clearer to define variables such as the ratio of the yaw-misaligned wind speed to the yaw-aligned wind speed called a wind speed ratio. Then, the goal of the NTF is to predict the wind speed ratio. In Figure 10 you could then plot the wind speed ratio predicted by the NTF removing any ambiguity. Similar variables could be defined for the wind vane bias or power ratio. These quantities could be defined in the methods section and used throughout for consistency and clarity.

- The titles of Sections 4.1 and 4.2 are not descriptive enough. Please use titles which better explain the content of the section such as "Estimating expected yaw-aligned operation using weighted neighbor averaging methods".

- The use of the term "stable" to mean a lack of spatial variability is confusing (L212, L333) since stability is commonly used to describe thermal stratification of the atmospheric boundary layer.

- The acronym CRMSE should be defined when first introduced (Table 5).

- Table 5 would be more informative if measures of variability and uncertainty (e.g., standard deviations, standard errors) were included. Is the difference between methods statistically significant?

- For conciseness and clarity, I prefer to have all subfigures individually lettered, and then to use the lettering in the captions (e.g. Figure 5) to refer to the subfigures. Also, in general all references to the figures in the text should have the figure number and the letter of the subfigure (e.g. L271).